# EXK-SC: A Semantic Communication Model Based on Information Framework Expansion and Knowledge Collision

**DOI:** 10.3390/e24121842

**Published:** 2022-12-17

**Authors:** Gangtao Xin, Pingyi Fan

**Affiliations:** 1Department of Electronic Engineering, Tsinghua University, Beijing 100084, China; 2Beijing National Research Center for Information Science and Technology, Tsinghua University, Beijing 100084, China

**Keywords:** semantic information theory, semantic communications, information theory, 6G, game theory

## Abstract

Semantic communication is not focused on improving the accuracy of transmitted symbols, but is concerned with expressing the expected meaning that the symbol sequence exactly carries. However, the measurement of semantic messages and their corresponding codebook generation are still open issues. Expansion, which integrates simple things into a complex system and even generates intelligence, is truly consistent with the evolution of the human language system. We apply this idea to the semantic communication system, quantifying semantic transmission by symbol sequences and investigating the semantic information system in a similar way as Shannon’s method for digital communication systems. This work is the first to discuss semantic expansion and knowledge collision in the semantic information framework. Some important theoretical results are presented, including the relationship between semantic expansion and the transmission information rate. We believe such a semantic information framework may provide a new paradigm for semantic communications, and semantic expansion and knowledge collision will be the cornerstone of semantic information theory.

## 1. Introduction and Overview

It is well known that Shannon’s information theory [1] answers two fundamental questions in digital communication theory [2], namely about the ultimate data compression rate (answer: the entropy *H*) and the ultimate reliable transmission rate of communication (answer: the channel capacity *C*). It also addresses technical implementation problems in digital communication systems, enabling the end user to receive the same symbols as the sender. However, with the ever-increasing demand for intelligent wireless communications, communication architecture is evolving from only focusing on transmitting symbols to the intelligent interconnection of everything [3]. In the 1950s, Weaver [4] discussed the semantic problem of communications, and categorized communications into the following three levels:

Level A. How accurately can the symbols of communication be transmitted? (The technical problem.)

Level B. How precisely do the transmitted symbols convey the desired meaning? (The semantic problem.)

Level C. How effectively does the received meaning affect conduct in the desired way? (The effectiveness problem.)

Recently, semantic information theory including semantic communications has attracted much attention. However, how to set up a reasonable and feasible measure of semantic messages is still an open problem, which may be the greatest challenge for the new developments of semantic communication (SC) systems. On the other hand, currently in 6G networks, intelligent interconnections of everything will bring a new paradigm of the communication mode with semantic interaction. In this regard, finding a way to quickly reflect the semantic processing of messages will become one more promising pathway to 6G intelligent networking systems.

This paper proposes a new communication system framework based on information framework expansion and knowledge collision. It extends Shannon’s theory of communication (Level A) to a theory of semantic communication (Level B). This work is initially influenced by Choi and Park [5], but it provides further important contributions, which are as follows:We generalize the work of Choi and Park from a single fixed semantic type to a dynamic expansion mode, associated with knowledge collision, which can reflect the asynchronous knowledge update processes between the sender and the receiver to some degree. In addition, it also takes into account the effect of channel noise in the new model.We present a new measure related to the semantic communication system based on the framework with semantic expansion and knowledge collision, called the Measure of Comprehension and Interpretation, which can be used to quantify the semantic entropy of discrete sources.We discuss the additional gains from semantic expansion and find the relationship between the semantic expansion and the transmission information rate. We demonstrate that knowledge matching of its asynchronous scaling up plays a key role in semantic communications.

As a primary work, the system proposed in this paper only focuses on discrete cases and makes some drastic simplifications. We assume that semantics are abstractly generated only from signals and knowledge, without considering other factors. In addition, semantic types, signals, responses, and knowledge instances (explained in Section 2) are represented by random variables. Moreover, we do not focus on the effectiveness problem (Level C), which is beyond the scope of this paper. However, we believe that these simplifications are necessary for us to focus on the “core” issues to clearly explain and set up connections within a general semantic information theory. To the best of our knowledge, this is the first study to explore the semantic expansion and the asynchronous scaling up of semantic knowledge. The insights gained from this work may be of great assistance to the development of future intelligent semantic communication systems and 6G.

The rest of this paper is organized as follows. Section 2 introduces the related work of semantic communications. It summarizes the key concepts and theoretical results of Shannon’s information theory and semantic information theory. In Section 3, we present the new generalized model of semantic communications from two aspects, one is an attempt to set up a close relationship with the Shannon communication model, and the other is to find a feasible modification of the model so that it can exactly reflect the quick surge of semantic communications from the user requirements. In Section 4, we lay out the implementation details of the simulation and discuss the experimental results. Finally, we conclude the paper in Section 5.

## 2. Related Work

In this section, we introduce the framework of semantic information theory, specifically the key concept, semantic entropy. We then go on to explore the works that we are concerned with, semantic communication as a signaling game with correlated knowledge bases.

### 2.1. Preliminaries

Although Shannon’s information did not address the semantic problem of communications, it provided important insights into the message-processing techniques associated with the focus of both the sender and the receiver’s attention. Thus, in this subsection, we briefly introduce the main concepts and theoretical results of Shannon’s information theory.

**Entropy.** It is a measure of the uncertainty of a random variable [2]. Let *X* be a discrete random variable with alphabet X and probability mass function p(x)=Pr{X=x},x∈X. The *entropy*
H(X) of *X* is defined as follows:(1)$$H\left(X\right)=-\sum_{x\in X}p\left(x\right)\log p\left(x\right).$$

**Mutual information.** It is a measure of the amount of information that one random variable contains about another. That is, mutual information can be seen as the reduction in the uncertainty of one random variable due to the knowledge of the other. Let us consider two random variables *X* and *Y* with a joint probability mass function p(x,y) and the marginal probability mass functions p(x) and p(y). The mutual information between *X* and *Y* is defined as follows:(2)$$I\left(X;Y\right)=\sum_{x\in X}\sum_{y\in Y}p\left(x,y\right)\frac{p(x,y)}{p(x)p(y)}.$$

**Channel capacity.** The channel capacity is the maximum amount of mutual information given the conditional transit probability from *X* to *Y*, p(y|x). It is defined by
(3)$$C=\max_{p(x)}I\left(X;Y\right),$$
where *X* and *Y* are the input and output of the channel, respectively.

**Source coding theorem.** As N→∞, *N* i.i.d. (independent identically distribution) random variables with entropy H(X) can be compressed into a little more than NH(X) bits to represent them completely, the information loss can be negligible in this case. Conversely, if they are compressed into fewer than NH(X) bits, there will be errors with a non-zero probability.

**Channel coding theorem.** For a discrete memoryless channel, all rates below capacity *C* are achievable. Conversely, any sequence of codes with a negligible error must obey the rule that its transmission rate *R* is not greater than the channel capacity, R≤C.

### 2.2. Semantic Information Theory

In the literature, there are some works that focus on the semantic information theory. Carnap and Bar-Hillel [6] were the first to propose the concept of semantic entropy, using logical probability rather than statistical probability to measure the semantic entropy of a sentence. For simplicity, it is necessary here to clarify that we use *p* and *m* to denote statistical probability and logical probability in this paper, respectively. The logical probability of a sentence is measured by the likelihood that the sentence is true in all possible situations [7]. Then, the semantic information of the message *e* is defined as follows:(4)$$H_{S}\left(e\right)=-\log_{2}\left(m\left(e\right)\right),$$
where m(e) is the logical probability of *e*. However, this metric led to a paradox that any fact has an infinite amount of information when it contradicts itself, i.e., $H_{S}\left(e\wedge \neg e\right)=\infty$. Floridi [8] solved the paradox in Carnap and Bar-Hillel’s proposal [6], adopting the relative distance of semantics to measure the amount of information.

Bao et al. [7] defined the semantic entropy of a message *x* as follows:(5)$$H_{S}\left(x\right)=-\log_{2}\left(m\left(x\right)\right),$$
where the logical probability of *x* is given by
(6)$$m\left(x\right)=\frac{\mu (W_{x})}{\mu (W)}=\frac{\sum_{w\in W,w⊧x}\mu \left(w\right)}{\sum_{w\in W}\mu \left(w\right)}.$$
*W* is the symbol set of a source, ⊧ is the proposition satisfaction relation, and $W_{x}$ is the set of models for *x*. In addition, μ is a probability measure, $\sum_{w\in W}\mu \left(w\right)=1$.

Besides the logical probability, there are some definitions of semantic entropy based on different backgrounds [9,10]. D’Alfonso [11] utilized the notion of truthlikeness to quantify semantic information. Kolchinsky and Wolpert [12] defined semantic entropy as the syntactic information that a physical system has about its environment, which is necessary for the system to maintain its own existence. Kountouris and Pappas [13] advocated for assessing and extracting the semantic value of data at three different granularity levels, namely the microscopic scale, mesoscopic scale, and macroscopic scale.

Analogous to Shannon’s information theory, some related theories at the semantic level, such as semantic channel capacity, semantic rate distortion, and information bottleneck [14], are also explored. Based on (Equation 5), Bao et al. further proposed the *semantic channel coding theorem* [7]. Moreover, Liu et al. [15] formulated rate distortion in semantic communication as follows:(7)$$R\left(D_{s},D_{w}\right)=\min I\left(W;\hat{X},\hat{W}\right)$$
where $D_{s}$ is the semantic distortion between source, *X*, and recovered information, $\hat{X}$, at the receiver. $D_{w}$ is the distortion between semantic representation, *W*, and received semantic representation, $\hat{W}$. In addition, some works [16,17] generalized it, leading to the development of semantic rate distortion theory.

In recent years, the fusion of semantic communication algorithms and learning theory [18,19,20,21] has driven the changes in communication architecture. Although these explorations did not fully open the door to semantic communication, they provided us with the motivation to move forward theoretically.

### 2.3. Semantic Communication as a Lewis Signaling Game with Knowledge Bases

Recently, Choi and Park [5] proposed an SC model based on the Lewis signaling game, which provided some meaningful results. Their work motivated us to conduct this study to some degree. Further, we make a generalization of this model by adding the scaling up update module of the knowledge instance at both the sender and the receiver, which may be asynchronous, and closer to real semantic communications in the era of intelligent communication, i.e., human to robots, robots to robots, machines to machines and cells to cells in bio-molecular communications, etc. Thus, we provide further details.

Suppose there is a semantic communication system where Alice is the sender and Bob is the receiver. Let T∈T denote the semantic type that includes semantic information or messages. Alice wishes to transmit *T* to Bob by sending a signal S∈S, and Bob chooses its response R∈R. Both Alice and Bob utilize their local knowledge bases $K_{A}$ and $K_{B}$, respectively. The semantic architecture can be described as follows:$$\begin{array}{cc} \; & K_{A}\;\;\;\;\;\;\;\;\;\;\;\;K_{B}\\ \; & \;↓\;\;\;\;\;\;\;\;\;\;\;\;\;\;\;↓\\ \; & T\;\to \;S\;\to \;R\;(=\hat{T}), \end{array}$$
where $K_{A}\in K_{A}$ and $K_{B}\in K_{B}$ are knowledge instances. If R=T, the communication process can be seen as successful. Moreover, the reward for one communication is defined as follows:(8)$$u=\left\{\begin{array}{c} 1,\;\mathrm{if}\;R=T;\\ 0,\;\mathrm{otherwise}. \end{array}\right.$$
Our object is to maximize the average reward. We use *the success rate of semantic agreement (SRSA)* to represent it.

The knowledge base can be regarded as the side information. Alice has her knowledge base $K_{A}$. The instance of Alice’s knowledge base at each time is $K_{A}\in K_{A}$, which affects the generation of semantic types. On the other hand, Bob has his knowledge base $K_{B}$ and its instance is $K_{B}\in K_{B}$. $K_{B}$ is the “core”, which is used to infer the intended message together with the received signal.

Based on these foundations, Choi and Park [5] derived some meaningful results. On the one hand, the similarity of the two knowledge instances plays a crucial role in SC. With limited bandwidth, the communication quality is regulated by $I(K_{A};K_{B}|S)$. On the other hand, the SRSA is dependent on the similarity of knowledge bases at both ends of the communication, the sender and the receiver. However, a close relationship with Shannon’s communication model was not established. In this work, we will construct a generalization of the model by adding the knowledge scaling up update module at both the sender and the receiver, which may be asynchronous, but it can closely reflect the quick surge of semantic communications in many potential applications.

## 3. Semantic Communication Models

In this section, we will present the generalized model of semantic communications from two different aspects, one is an attempt to set up a close relationship with the Shannon communication model, and the other is to find a feasible modification of the model so that it can exactly reflect the quick surge of semantic communications from the user requirements. In this regard, we first set up a basic model, and then extend it by adding the knowledge base update functional module with information framework expansion and knowledge collision.

### 3.1. Motivation

Shannon’s communication system is concerned with the accurate transmission of symbols over channels. However, it does not take into account the differences in the knowledge backgrounds of the two parties, which affect whether the communication can convey the desired meaning. In other words, the matching of knowledge can also be treated as another special channel, although it may not actually exist in an explicit form. In this work, we denote it as a virtual channel. We want to fill this gap and propose an intelligent communication system with the coexistence of the real channel and virtual channel, making it easier to formulate the information framework. Thus, we consider not only the physical channel for symbol transmission but also the virtual channel for the transfer of knowledge between two parties. Moreover, the knowledge of both parties is also evolving, so we take into account the asynchronous scaling up updates of knowledge and semantic expansion.

### 3.2. Basic Model

Let us first observe the *channel*, which is the physical medium of message exchange, and plays a key role in the model setup of a complete semantic communication system. A *communication channel* is a system in which the output depends probabilistically on its input [2]. For simplification, we still use Alice and Bob as the two parties participating in the communication. Suppose Alice and Bob are at opposite ends of a memoryless channel, which can transmit physical signals. The channel is said to be memoryless if the probability distribution of the output depends only on the input at that time and it is conditionally independent of previous inputs or outputs. That is, the output of the channel is only related to the input at the current moment and no feedback processing is considered here.

We use *S* and $\hat{S}$ to represent the input and output of the channel. Under the interference of the noise, the signal *S* becomes $\hat{S}$ through channel transmission. Specifically, Alice observes the input *S* of the channel, and Bob observes the output $\hat{S}$. Let $K_{A}$ and $K_{B}$ represent the knowledge instance possessed by Alice and Bob, respectively. On the one hand, Alice encodes *S* as *T* with $K_{A}$. This means that Alice uses the knowledge instance $K_{A}$ to process the signal *S*, resulting in the semantic type *T*, i.e., $(S,K_{A})\to T$. On the other hand, Bob decodes $\hat{S}$ into the response *R* with $K_{B}$, which can be expressed as $(\hat{S},K_{B})\to R$. The architecture of this process can be expressed as follows:(9)$$\begin{array}{cc} T\;\leftarrow \; & S\;\;\;\overset{\left(a\right)}{⟶}\;\;\;\hat{S}\;\to \;R\;\left(=\hat{T}\right)\\ \; & \;↑\;\;\;\;\;\;\;\;\;\;\;\;↑\\ \; & K_{A}\;\;\;\overset{\left(b\right)}{⤏}\;\;\;K_{B} \end{array}$$

We denote the process $(S,K_{A})\to T$ as semantic encoding and $\left(\hat{S},K_{B}\right)\to \hat{T}$ as semantic decoding. They indicate the understanding of meaning at the semantic level by both parties. The definitions of symbols are summarized as follows:*Semantic Types*: $T=t_{k},k=1,\ldots ,\left|T\right|$, is a random variable that is generated by Alice.*Signals*: $S=s_{l},l=1,\ldots ,\left|S\right|$, is a signal that Alice sends to Bob.*Responses*: $R=r_{n},n=1,\ldots ,\left|R\right|$, is a response that Bob chooses.*Knowledge instances*: $K_{A}\in K_{A}$ and $K_{B}\in K_{B}$ represent the knowledge instance used by Alice and Bob during semantic encoding and decoding, respectively.

**Lemma 1.** 
*Under the knowledge instances $K_{A}$ and $K_{B}$, the mutual information between T and $\hat{T}$ is obtained as follows:*

(10)
$$I\left(T;\hat{T}\right)=I\left(S;\hat{S}\right)+I\left(K_{A};K_{B}|S,\hat{S}\right)+I\left(S;K_{B}|\hat{S}\right)+I\left(\hat{S};K_{A}|S\right)$$

*where $I(K_{A};K_{B}|S,\hat{S})$, $I(S;K_{B}|\hat{S})$ and $I(\hat{S};K_{A}|S)$ represent the conditional mutual information.*


**Proof.** Since $(S,K_{A})\to T$ and $\left(\hat{S},K_{B}\right)\to \hat{T}$, we obtain $H\left(T\right)=H\left(K_{A},S\right)$ and $H\left(\hat{T}\right)=H\left(K_{B},\hat{S}\right)$. With the characteristics of entropy and mutual information, it follows that
(11)$$\begin{array}{cc} I(T;\hat{T}) & =H\left(T\right)-H\left(T|\hat{T}\right) \end{array}$$
(12)$$\begin{array}{cc} \; & =H\left(K_{A},S\right)-H\left(K_{A},S|K_{B},\hat{S}\right) \end{array}$$
(13)$$\begin{array}{cc} \; & =H\left(S\right)+H\left(K_{A}|S\right)-H\left(S|K_{B},\hat{S}\right)-H\left(K_{A}|K_{B},S,\hat{S}\right) \end{array}$$
(14)$$\begin{array}{cc} \; & =H\left(S\right)-H\left(S|K_{B},\hat{S}\right)+H\left(K_{A}|S\right)-H\left(K_{A}|K_{B},S,\hat{S}\right) \end{array}$$
(15)$$\begin{array}{cc} \; & =I\left(S;K_{B},\hat{S}\right)+I\left(K_{A};K_{B},\hat{S}|S\right) \end{array}$$
(16)$$\begin{array}{cc} \; & =I\left(S;\hat{S}\right)+I\left(K_{A};K_{B}|S,\hat{S}\right)+I\left(S;K_{B}|\hat{S}\right)+I\left(\hat{S};K_{A}|S\right), \end{array}$$
which completes the proof. □

The mutual information between *T* and $\hat{T}$ reflects the effectiveness of communication. It indicates the highest rate in bits per channel use at which information can be sent with an arbitrarily low probability of error. We note that $I(T;\hat{T})$ consists of the following three terms:$I(S;\hat{S})$, the mutual information between the input and output of the channel. It corresponds to the *channel capacity* in Shannon’s information theory.$I(K_{A};K_{B}|S,\hat{S})$, the mutual information between $K_{A}$ and $K_{B}$ given *S* and $\hat{S}$. It indicates the amount of information that two knowledge instances contain about each other when signals are known.$I\left(S;K_{B}|\hat{S}\right)+I\left(\hat{S};K_{A}|S\right)$, the conditional mutual information between *S* and $K_{B}$, $\hat{S}$ and $K_{A}$.

From Equation (Equation 10), we know that if Bob wants to fully understand what Alice means, the communication process needs to meet two conditions, one is the accuracy of the received symbols during channel transmission, and the other is the matching degree of the knowledge instances of both parties. In other words, (i) The transmission $S\to \hat{S}$ should be reliable. It indicates the effect of channel noise on signal transmission. Moreover, the physical carrier of this process actually exists, which we call an *explicit channel*. (ii) The two knowledge instances should be similar. Although there is no actual transmission between $K_{A}$ and $K_{B}$, we assume that there is a virtual channel that reflects the probabilistic relationship between knowledge instances, which we call an *implicit channel*. The *explicit channel* and *implicit channel* together form the transmission medium of a communication system. Specifically, the characteristics of the explicit channel determine the value of the first term in Equation (Equation 10), and the implicit channel determines the second term. In addition, the last term is affected by both explicit and implicit channels.

If the communication system has only the explicit channel, it degenerates to the Shannon case, and $I\left(T;\hat{T}\right)=I\left(S;\hat{S}\right)$. Furthermore, we are interested in the following three special cases (demonstrated in Appendix A).
**The explicit channel is noiseless**, i.e., $S=\hat{S}$. In this setting, the Formula (Equation 10) can be simplified to
(17)$$I\left(T;\hat{T}\right)=H\left(S\right)+I\left(K_{A};K_{B}|S\right),$$
which is consistent with the result in [5]. The mutual information between *T* and $\hat{T}$ is subjected to the entropy of the signal *S* and the conditional mutual information between $K_{A}$ and $K_{B}$ given *S*. In semantic communications or emergent communications, the signal *S* is usually limited. According to Equation (Equation 17), one can increase the conditional mutual information between $K_{A}$ and $K_{B}$ to solve the problem from another perspective. In other words, in the case of limited technical bandwidth, the SC performance can be improved by increasing the similarity between two knowledge instances.**The implicit channel is noiseless**, i.e., $K_{A}=K_{B}$. This means that Alice and Bob have the same knowledge instance, so the communication performance will not be affected by the difference in the background of both parties. $I(T;\hat{T})$ can be expressed as follows:
(18)$$\begin{array}{cc} \; & I\left(T;\hat{T}\right)=I\left(S;\hat{S}\right)+H\left(K_{A}\right)-I\left(K_{A};S;\hat{S}\right) \end{array}$$
(19)$$\begin{array}{cc} \; & I\left(T;\hat{T}\right)=I\left(S;\hat{S}\right)+H\left(K_{B}\right)-I\left(K_{B};S;\hat{S}\right), \end{array}$$
where $I\left(K_{A};S;\hat{S}\right)=I\left(K_{A};S\right)-I\left(K_{A};S|\hat{S}\right)$ is the mutual information between $K_{A}$, *S* and $\hat{S}$. Even when the implicit channel is noiseless, $I(T;\hat{T})$ is not exactly equal to $I(S;\hat{S})$, and we cannot ignore the influence of knowledge in SC. Compared to Shannon’s channel capacity, it adds a term, regulated by the relationship between the knowledge instance and the signal.

**Corollary 1.** 
*If the implicit channel is noiseless, the mutual information between T and $\hat{T}$ satisfies*

(20)
$$I\left(S;\hat{S}\right)\overset{\left(a\right)}{\leq }I\left(T;\hat{T}\right)\overset{\left(b\right)}{\leq }I\left(S;\hat{S}\right)+H\left(K_{A}\right).$$

*Moreover, if $K_{A}$ is a function of S and $\hat{S}$, the left equation (a) holds; if $K_{A}$ is independent of the signal S or $\hat{S}$, the right equation (b) holds.*


**Proof.** The non-negativity of mutual information, $I(K_{A};S;\hat{S})\geq 0$, means that $I(T;\hat{T})$ can be upper bounded by $I\left(S;\hat{S}\right)+H\left(K_{A}\right)$. Because the mutual information is lower than the entropy, we obtain $H\left(K_{A}\right)-I\left(K_{A};S;\hat{S}\right)\geq 0$. By combining these results, we obtain
(21)$$I\left(S;\hat{S}\right)\leq I\left(T;\hat{T}\right)\leq I\left(S;\hat{S}\right)+H\left(K_{A}\right),$$
which completes the proof of the bound of $I(T;\hat{T})$. □

III.**Both explicit and implicit channels are noiseless**, i.e., $S=\hat{S},K_{A}=K_{B}$. In this setting, the mutual information between *T* and $\hat{T}$ equals the entropy of *T* or $\hat{T}$,
(22)$$I\left(T;\hat{T}\right)=H\left(T\right)=H\left(\hat{T}\right).$$Equation (Equation 22) indicates that Bob can perfectly understand the meaning of Alice when channels are noiseless. That is, no information is lost.

The following result shows the bounds of SRSA, constrained by the characteristics of the explicit and implicit channels.

**Lemma 2.** 
*The SRSA satisfies*

(23)
$$\mathrm{SRSA}=\mathrm{Pr}\left\{T=\hat{T}\right\}\leq 1-\frac{H(K_{A},S|K_{B},\hat{S})-1}{\log|T|}$$



**Proof.** $\left(K_{B},\hat{S}\right)$ can be seen as an estimator for $\left(K_{A},S\right)$. Let $P_{e}=$ Pr$\left\{T\neq \hat{T}\right\}$; then, with Fano’s Inequality, we obtain
(24)$$P_{e}\geq \frac{H(K_{A},S|K_{B},\hat{S})-1}{\log|T|}$$ Since SRSA $=1-P_{e}$, we can obtain Equation (Equation 23), which completes the proof. □

### 3.3. EXK-SC Model

Let $S_{1}$ and $S_{2}$ denote the input signals of the explicit channel. ${\hat{S}}_{1}$ and ${\hat{S}}_{2}$ are their corresponding output signals. Similarly, $K_{A}^{1}$ and $K_{A}^{2}$ represent Alice’s knowledge instances. Alice encodes $S_{1}$ as $T_{1}$ with $K_{A}^{1}$, and $S_{2}$ as $T_{2}$ with $K_{A}^{2}$. On the other hand, Bob uses $K_{B}^{1}$ and $K_{B}^{2}$, decoding ${\hat{S}}_{1}$ and ${\hat{S}}_{2}$ into ${\hat{T}}_{1}$ and ${\hat{T}}_{2}$, respectively. In the basic model, the encoding and decoding processes of the two transmissions are
(25)$$\begin{array}{cc} S_{1},K_{A}^{1} & \to T_{1}\;\;\;\;{\hat{S}}_{1},K_{B}^{1}\to {\hat{T}}_{1} \end{array}$$
(26)$$\begin{array}{cc} S_{2},K_{A}^{2} & \to T_{2}\;\;\;\;{\hat{S}}_{2},K_{B}^{2}\to {\hat{T}}_{2} \end{array}$$

It is well known that the integration of lots of simple things can be expanded to a complex system and can even generate intelligence. Such an expansion process is consistent with the evolution of the human language system. Based on a similar idea, we extend the basic model proposed in Section 2.3 using expansion. In this context, Alice wishes to send an expansion of multiple signals to Bob. We first consider the case of two signals, which can be generalized to more instances. Now, what Alice sends is expanded from $S=S_{1}$ to $S=S_{1}⊕T_{2}$. We use ⊕ to denote semantic expansion. Specifically, the expansion of signals only represents their semantic combination. For instance, ‘*Shannon published a paper*’ expands to ‘*Shannon published a paper in The Bell system technical journal* [1]’.

As known, expansion often implies collision and fusion. Similarly, the expansion of signals corresponds to the collision of knowledge bases in this work. For Alice, we use $K_{A}=K_{A}^{1}\overset{\alpha }{⊙}K_{A}^{2}$ to represent the collision process, where ⊙ denotes the collision and α is the *collision factor*. Specifically, α is between 0 and 1, determined by the task. The collision factor reflects the role of $K_{A}^{2}$ compared to $K_{A}^{1}$ when the collision occurs. It can also be understood as the relative proportion of the contribution to the newly generated knowledge instance. The process of collision represents the knowledge scaling up updates, which may be synchronous or asynchronous. Similarly, we use $K_{B}=K_{B}^{1}\overset{\beta }{⊙}K_{B}^{2}$ to represent the knowledge collision of Bob, where β is a collision factor. The expansion and collision process can proceed continuously as follows: (27)$$\begin{array}{cc} \; & S=S_{1}⊕S_{2}⊕S_{3}\cdots ⊕S_{n} \end{array}$$(28)$$\begin{array}{cc} \; & K_{A}=K_{A}^{1}\overset{\alpha_{1}}{⊙}K_{A}^{2}\overset{\alpha_{2}}{⊙}K_{A}^{3}\cdots \overset{\alpha_{n-1}}{⊙}K_{A}^{n} \end{array}$$(29)$$\begin{array}{cc} \; & K_{B}=K_{B}^{1}\overset{\beta }{⊙}K_{B}^{2}\overset{\beta_{2}}{⊙}K_{B}^{3}\cdots \overset{\beta_{n-1}}{⊙}K_{B}^{n}. \end{array}$$

Without loss of generality, the one-step expansion architecture of semantic communications is described as follows:(30)$$\begin{array}{cc} \; & S\;\;\;\;\;\;\overset{\left(a\right)}{⟶}\;\;\;\;\;\;\hat{S}\\ \; & ⇑\;\;\;\;\;\;\;\;\;\;\;\;\;\;\;\;\;\;\;⇓\\ T\;\leftarrow \;S_{1} & ⊕\;\;S_{2}\;\;\;\;\;\;\;\;\;\;\hat{S_{1}}\;⊕\;\hat{S_{2}}\;\to \;R\;\left(=\hat{T}\right)\\ \; & ↑\left(c\right)\;\;\;\;\;\;\;\;\;\;\;\;\;\;↑\left(d\right)\\ K_{A}^{1} & \overset{\alpha }{⊙}\;\;K_{A}^{2}\;\;\;\overset{\left(b\right)}{⤏}\;\;K_{B}^{1}\;\overset{\beta }{⊙}\;K_{B}^{2} \end{array}$$

We named it EXK-SC. Moreover, $H(S_{1}⊕S_{2},K_{A}^{1}\overset{\alpha }{⊙}K_{A}^{2})$ is called the Measure of Comprehension and Interpretation (MCI), which reflects the generation and evolution of semantics. It should be noted that *T* is not a simple logic combination of $T_{1}$ and $T_{2}$, i.e., $T\neq T_{1}⊕T_{2}$. For example, $T_{1}$ is ‘*Apple Inc.*’, it is a company and $S_{1}$ can be ‘*Apple*’. $T_{2}$ is ‘*the thirteenth generation*’, it is a number and $S_{2}$ can be ‘*thirteen*’. However, their collision may give rise to a new word called ‘*iphone*’, which is a mobile communication product. In particular, *T* reflects the result of *S* under the influence of knowledge collision.

Based on these definitions above, we obtain some new results.

**Lemma 3.** 
*The mutual information between T and $\hat{T}$ is given by*

(31)
$$I\left(T;\hat{T}\right)=I\left(S;\hat{S}\right)+I\left(K_{A}^{1}\overset{\alpha }{⊙}K_{A}^{2};K_{B}^{1}\overset{\beta }{⊙}K_{B}^{2}|S,\hat{S}\right)+I\left(S;K_{B}^{1}\overset{\beta }{⊙}K_{B}^{2}|\hat{S}\right)+I\left(\hat{S};K_{A}^{1}\overset{\alpha }{⊙}K_{A}^{2}|S\right).$$



**Proof.** It is similar to that of Lemma 1, we omit the proof here. □

Equation (Equation 31) indicates that besides the characteristics of explicit and implicit channels, the relationship between α and β also affects the performance of communication.

**Lemma 4.** 
*When the explicit channel is noiseless, the gain brought by semantic expansion is given by*

(32)
$$\begin{array}{c} I\left(T;\hat{T}\right)-I\left(T_{1};{\hat{T}}_{1}\right)=\left(1+\gamma \right)\left(H(S)-H\left(S_{1}\right)\right), \end{array}$$

*where*

(33)
$$\gamma =\frac{I\left(K_{A}^{1}\overset{\alpha }{⊙}K_{A}^{2};K_{B}^{1}\overset{\beta }{⊙}K_{B}^{2}|S_{1}⊕S_{2}\right)-I\left(K_{A}^{1};K_{B}^{1}|S_{1}\right)}{H(S_{1}⊕S_{2})-H\left(S_{1}\right)}.$$



**Proof.** We note that
(34)$$I\left(T_{1};{\hat{T}}_{1}\right)=H\left(S_{1}\right)+I\left(K_{A}^{1};K_{B}^{1}|S_{1}\right).$$ Then,
(35)$$\begin{array}{cc} \; & \frac{I\left(T;\hat{T}\right)-I\left(T_{1};{\hat{T}}_{1}\right)}{H(S)-H(S_{1})} \end{array}$$
(36)$$\begin{array}{cc} \; & =\frac{H\left(S\right)+I\left(K_{A}^{1}\overset{\alpha }{⊙}K_{A}^{2};K_{B}^{1}\overset{\beta }{⊙}K_{B}^{2}|S\right)-H\left(S_{1}\right)-I\left(K_{A}^{1};K_{B}^{1}|S_{1}\right)}{H(S_{1}⊕S_{2})-H\left(S_{1}\right)} \end{array}$$
(37)$$\begin{array}{cc} \; & =1+\frac{I\left(K_{A}^{1}\overset{\alpha }{⊙}K_{A}^{2};K_{B}^{1}\overset{\beta }{⊙}K_{B}^{2}|S\right)-I\left(K_{A}^{1};K_{B}^{1}|S_{1}\right)}{H(S_{1}⊕S_{2})-H\left(S_{1}\right)} \end{array}$$
(38)$$\begin{array}{cc} \; & =1+\gamma . \end{array}$$ We have
(39)$$\begin{array}{c} I\left(T;\hat{T}\right)-I\left(T_{1};{\hat{T}}_{1}\right)=\left(1+\gamma \right)\left(H(S)-H\left(S_{1}\right)\right), \end{array}$$
which completes the proof. □

Lemma 4 shows that the semantic enhancement, $I\left(T;\hat{T}\right)-I\left(T_{1};{\hat{T}}_{1}\right)$, is determined not only by the knowledge scaling up but also by the cost of the information rate over the explicit physical channel, $H\left(S\right)-H\left(S_{1}\right)$. It exactly illustrates the relationship between the semantic expansion and the transmission information rate.

**Lemma 5.** 
*When the explicit channel is noiseless, the mutual information between T and $\hat{T}$ is bounded by*

(40)
$$\frac{1}{2}\left(H\left(S_{1}\right)+H\left(S_{2}\right)\right)\leq I\left(T;\hat{T}\right)\leq H\left(K_{A}^{1}\overset{\alpha }{⊙}K_{A}^{2}\right)+H\left(S\right)$$



**Proof.** Since expansion would create more possibilities, leading to an increase in uncertainty, we can obtain
(41)$$H\left(S_{1}⊕S_{2}\right)\geq H\left(S_{1}\right)\;\;\;\;H\left(S_{1}⊕S_{2}\right)\geq H\left(S_{2}\right).$$ Because of the *non-negativity of entropy* and *conditioning reduces entropy*, Equation (Equation 40) can be derived directly. We omit the proof. □

Lemma 5 shows the bounds of the semantic communication rate in the perfect explicit channel mode. The upper bound is composed of the entropy rate over the explicit channel and the collision of knowledge instances of the sender, which is reasonable, as expected.

## 4. Experiment and Numerical Results

In this section, we use SRSA to measure the performance of SC, especially the impact that asynchronous knowledge scaling up has on the system.

**Basic Model.** We use Q-learning in [22] to complete semantic encoding and decoding, which may reflect the continuous semantic learning process. Let |S|=M=2, $|K_{A}\left|=\right|K_{B}|=L=2$, $\left|T\right|=L^{2}M^{2}$. In addition, *S*, $K_{A}$ and $K_{B}$ are uniformly distributed. We would like to discuss the impact of the characteristics of the explicit channel and implicit channel on SC. We use the binary symmetric channel (BSC) to denote the explicit channel. The BSC is shown in Figure 1, which shows the probabilistic relationship between *S* and $\hat{S}$, with the error probability $\epsilon_{1}$.

For the implicit channel, we assume the correlation between $K_{A}$ and $K_{B}$ satisfies
$$K_{B}=\left\{\begin{array}{cc} K_{A}, & \mathrm{with}\;\mathrm{probability}\;1-\epsilon_{2}\\ U, & \mathrm{with}\;\mathrm{probability}\;\epsilon_{2}, \end{array}\right.$$
where U∼Unif{1,L} is an independent random variable. $\epsilon_{2}$ illustrates the discrepancy between the knowledge bases at both parties of communications, $0<\epsilon_{2}<1$. In Figure 2, we show the SRSA with $\epsilon_{1}\in$[0,0.5] and $\epsilon_{2}\in$[0,0.5]. When $\epsilon_{1}=\epsilon_{2}=0$, SRSA reaches the maximum 1. As $\epsilon_{1}$ and $\epsilon_{2}$ increase, SASA will decrease in both directions. This indicates that two channels jointly determine the quality of communication. For high-quality SC, it is necessary to meet the condition that $\hat{S}=S$ and $K_{B}=K_{A}$ with a high probability.

In addition, we discuss the impact of explicit and implicit channels on SC, respectively. We take the channel error $\epsilon_{1}$ and $\epsilon_{2}$ as the horizontal axis, and the SRSA entering the stable region as the vertical axis. Figure 3a shows the results of the explicit channel. As $\epsilon_{1}$ increases, the mean value of SRSA first decreases and then has a gradual increase. When $\epsilon_{1}$ = 0 or 1, the communication performance is the best and the mean is close to 1. Moreover, it falls to a low point when $\epsilon_{1}=0.5$. On the other hand, the trend of variance is completely opposite to that of the mean. These results are consistent with the description of channels in Shannon’s information theory. Figure 3b shows the results of the implicit channel. As $\epsilon_{2}$ increases, the mean value of SRSA keeps decreasing, and the variance increases. It reflects the impact of knowledge misalignment on SC.

**EXK-SC Model.** We want to explore how semantic communication quality varies in the context of different relationships between the receiver and the sender’s knowledge instance. For simplicity, we assume that the explicit channel is noiseless, so we can focus on knowledge updates. That is, SRSA varies with the relationship of $K_{B}^{1},K_{B}^{2},\beta$, and $K_{A}^{1},K_{A}^{2},\alpha$. We categorize it into four cases.
Case I: $K_{B}^{1}=K_{A}^{1}$, $K_{B}^{2}=K_{A}^{2}$ and β=α. The implicit channel is noiseless. Bob has the same asynchronous knowledge scaling-up updates mode as Alice. That is, the receiver has all the knowledge of the sender.Case II: $K_{B}^{1}\neq K_{A}^{1}$ (with error probability 0.5), $K_{B}^{2}=K_{A}^{2}$ and β=α. The receiver has partial knowledge of the sender.Case III: $K_{B}^{1}=K_{A}^{1}$, $K_{B}^{2}\neq K_{A}^{2}$ (with error probability 0.5) and β=α. The receiver has partial knowledge of the sender.Case IV: $K_{B}^{1}=K_{A}^{1}$, $K_{B}^{2}=K_{A}^{2}$ and $\beta =\frac{1}{2}\alpha$. The collision factors are not equal to this.

The symbols $K_{A}^{1},K_{A}^{2},\alpha ,K_{B}^{1},K_{B}^{2},\beta$ are consistent with the definitions in Section 3.3. Figure 4 illustrates the simulation results of SRSA with the number of runs, where Figure 4a is the mean and Figure 4b is the variance. In all four cases, the mean of SRSA gradually converges to a stable region. Case I is close to 1, which implies that when the receiver has all the knowledge of the sender, it can express exactly the same semantics as the sender. In other words, Bob has the same asynchronous knowledge scaling-up updates mode as Alice, which facilitates the success of SC. The curves of case II and III are almost the same, but the value of case II is always higher than that of case III. This is as expected because $K_{A}^{1}$ and $K_{B}^{1}$ play a leading role in the collision compared to $K_{A}^{2}$ and $K_{B}^{2}$. Case IV reflects the learning of SRSA when Bob and Alice have different collision factors. The value of case IV is higher than that of II and III, which indicates that the collision factor plays a smaller role than the knowledge itself in SC. These results also show that learning can improve SC quality with only partial background knowledge; however, there is an upper bound. On the other hand, the variance of case I continues decreasing, and the other three cases also stabilize after peaking quickly. This further suggests that learning can evolve continuously with full knowledge, but there is an upper limit to it only with partial or no knowledge. This can be treated as the cost resulting from the imperfect knowledge base.

## 5. Conclusions

In this paper, we presented the concept of semantic expansion and knowledge collision in SC. It represents the combination and superposition of information by the sender. Based on semantic expansion, we further proposed a semantic communication system called EXK-SC. Moreover, semantic expansion corresponds to knowledge collision, which provides the possibility for the evolution and upgrading of communication systems. On the other hand, we reached some conclusions for semantic information theory in the context of asynchronous scaling up updates of knowledge, while obtaining some bounds for SC. Specifically, the receiver’s understanding of the knowledge collision and updates determines the effectiveness of semantic communication.

It should be pointed out that in the near future, task-oriented semantic communication designs under this new framework may emerge as an interesting topic. Another topic of interest is how to set up the type class methods to provide more insights on the knowledge base formulation and the system design, which requires further study. Semantic communication is evolving towards intelligence. The insights gained from this work may be of assistance to the development of future semantic communication systems and 6G. It is also expected to pave the way for the design of next-generation real-time data networking and provide the foundational technology for a plethora of socially useful services, including autonomous transportation, consumer robotics, VR/AR, and the metaverse.

## Figures and Tables

**Figure 1 entropy-24-01842-f001:**
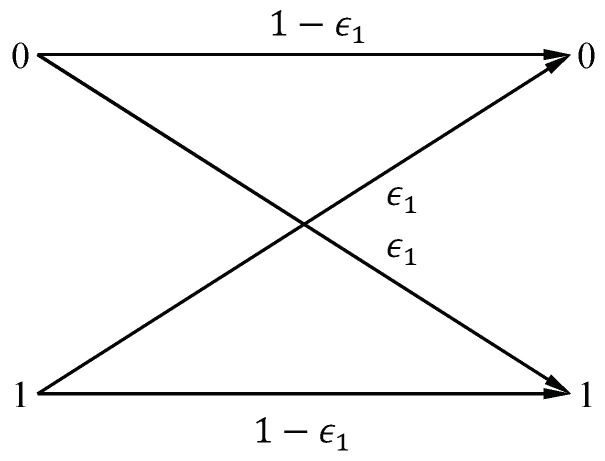
Binary symmetric channel.

**Figure 2 entropy-24-01842-f002:**
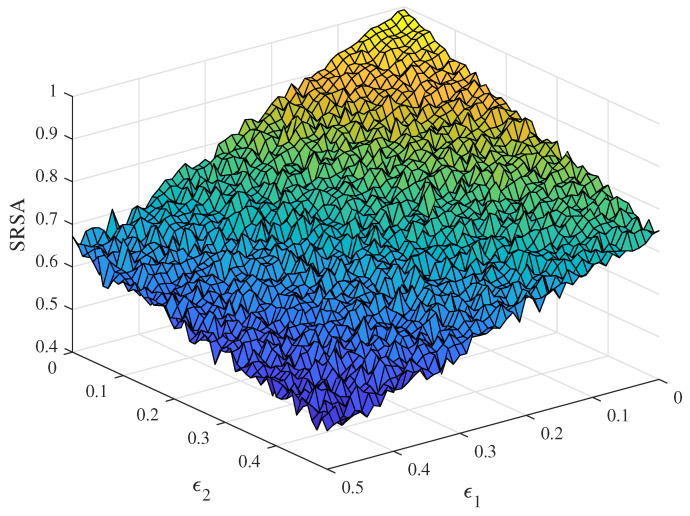
The simulation results of SRSA with the explicit channel error probability $\epsilon_{1}$ and the implicit channel error probability $\epsilon_{2}$.

**Figure 3 entropy-24-01842-f003:**
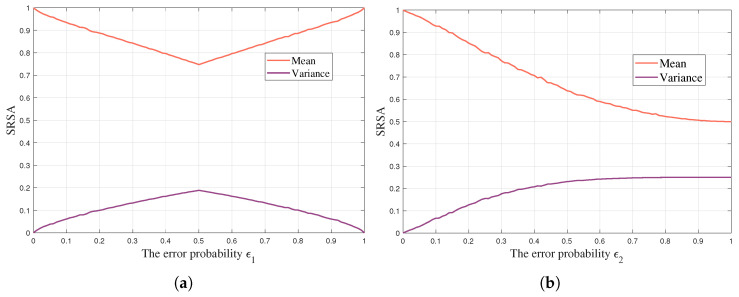
A comparison of the impact of explicit and implicit channels on SC. (**a**) Explicit channel. (**b**) Implicit channel.

**Figure 4 entropy-24-01842-f004:**
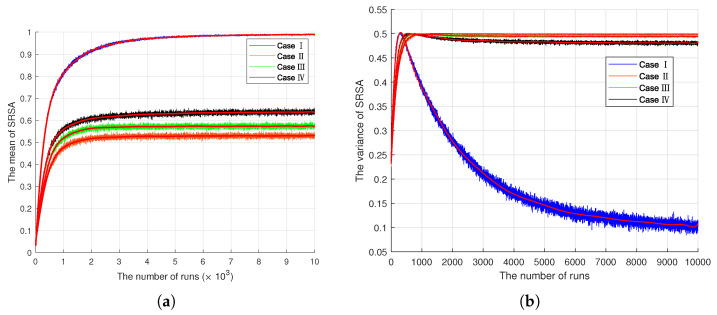
The simulation results of SRSA with the number of runs. We divide it into four cases. Case I: $K_{B}^{1}=K_{A}^{1}$, $K_{B}^{2}=K_{A}^{2}$ and β=α; II: $K_{B}^{1}\neq K_{A}^{1}$ (with an error probability of 0.5), $K_{B}^{2}=K_{A}^{2}$ and β=α; III: $K_{B}^{1}=K_{A}^{1}$, $K_{B}^{2}\neq K_{A}^{2}$ (with an error probability of 0.5) and β=α; IV: $K_{B}^{1}=K_{A}^{1}$, $K_{B}^{2}=K_{A}^{2}$ and $\beta =\frac{1}{2}\alpha$. (**a**) The mean of SRSA. (**b**) The variance of SRSA.

## Data Availability

Not applicable.

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
