# Peer review of "EXK-SC: A Semantic Communication Model Based on Information Framework Expansion and Knowledge Collision"

_entropy, 2022, doi:10.3390/e24121842_

Round 1
Reviewer 1 Report
In this paper, the authors propose a semantic communication model where encoding/decoding is performed under side information at both sides (named knowledge bases here). Extending the results of [5], the authors are trying to put some information theoretical modeling into the semantic communication problem, which is modeled as a Lewis signaling game.
Although the paper has some merit, there are several parts of the paper that need to be revised and clarified.
The concept of knowledge bases is key in this work. How calligraphic K_A and K_B are obtained or generated? Are they time-varying or evolving (or adapting based on the outcome/reward of the communication)? Furthermore, the authors should discuss the impact of having perfect common knowledge bases at both ends and the impact of the misalignment or dissimilarity between Ks on the overall performance.
The derivations especially for the explicit channel are not that new or unexpected, these are standard textbook extensions of Shannon's theory, treating knowledge and types that way is not fundamentally different on what Shannon theory can do. However, a key issue with that technical part is discussion on the operational meaning of these mutual information expressions, and most importantly a rigorous treatment of the sequences (typical) and achievability.
The simulation results show how SRSA behaves but comparison with other prior work or baseline is missing and what is also missing is a key plot where we can see the real (and significant) gains of this semantic approach (and EXP-SC) as compared to standard models or other SC approaches.
The way sentences are treated is as accumulation of symbols, so basically sequences of symbols, which combined make a work, which accumulated and combined with the knowledge base realization can be decoded at the receiver. This is a very simplistic view of words/sentences (and kind of old-fashioned) and cannot model/represent actual communication and meaning (not to mention that they fail to capture irony, metaphorical sentences, etc.). The authors should check on better representations (compositionality, distributional approaches, etc.) and improve their models.
Why the encoding/decoding is performed based on a realization of the Knowledge base (KB) and not the whole KB (distribution)? Although this simplify the model and basically make it trivial to reuse standard information theoretic tools, it oversimpifies the problem, as KB are richer than simple realizations of a random variable. This needs further work and elaboration.
Minor comments:
- The authors of [5] are mentioned as Jinho and Jihong in the paper (in many places) - however these are their first names. The convention is to use the family names, and this is has been done in all other references with names in the text.
- p. 4 [15] only the last author is mentioned and not the first authors. Moreover, [15] has been generalized (and corrected) by other papers, including https://arxiv.org/abs/2204.06049 and https://ieeexplore.ieee.org/document/9834593.
- Eq. (1) define the domain of x (\cal{X})
- EXP-SC is an acronym not properly defined or motivated, the EXP term.
A final minor comment, English usage and syntax could be improved; the authors are advised to have their paper read by a native speaker.
Reviewer 2 Report
The topic of this paper is interesting and in general it is well organized. The main reviewer concerns include,
· In the introduction is mentioned “makes some drastic simplifications”. The Reviewer suggest to point out the main simplification.
· Perhaps the Related Work section is too large. See if you can reduce it.
· Avoid this kind of sentences “For instance, ’Carol published a paper’ expands to ’Carol published a paper in IEEE Communications Letters’. Just use the references.
· How eq(30) is obtained?
· Is it possible to compare the proposed solution with others?
· Improve the English (do not use past in the abstract).
Round 2
Reviewer 2 Report
The authors have satisfactory addressed my concerns.